# Functional Characterization of the Wheat Macrophage Migration Inhibitory Factor TaMIF1 in Wheat-Stripe Rust (*Puccinia striiformis*) Interaction

**DOI:** 10.3390/biology10090878

**Published:** 2021-09-07

**Authors:** Mengxin Zhao, Qing Chang, Yueni Liu, Peng Sang, Zhensheng Kang, Xiaojie Wang

**Affiliations:** 1State Key Laboratory of Crop Stress Biology for Arid Areas, College of Plant Protection, Northwest A&F University, Yangling, Xianyang 712100, China; zhaomengxin@nwsuaf.edu.cn (M.Z.); ynn0322@163.com (Y.L.); sangpeng@nwsuaf.edu.cn (P.S.); 2Bio-Agriculture Institute of Shaanxi, Xi’an 710043, China; changq@xab.ac.cn

**Keywords:** wheat (*Triticum aestivum*), macrophage migration inhibitory factor, stripe rust susceptibility, reactive oxygen species

## Abstract

**Simple Summary:**

There have been many breakthroughs in MIF function and mechanism investigation in vertebrates, but it has rarely been studied in plants. Here, we aimed to characterize the function of MIF in wheat and its potential role in Wheat-Stripe rust interaction. We showed that wheat MIF has some similarities with that MIF in vertebrates, such as subcellular localization in both the cytosol and nuclei, as well as significant tautomerase activity, and both can inhibit Bax-induced programmed cell death. In the wheat–*Pst* interaction, *Ta*MIF1 is upregulated during *Pst* infection. Silencing *TaMIF1* decreased *Pst* infection of wheat tissues, and the accumulation of ROS was increased in *TaMIF1-silenced* wheat leaves, which hinted that *Ta*MIF1 mainly modulates the ROS signaling and then alters the subsequent immune responses. The function characterization of *Ta*MIF1 provides significant insight into the role of MIFs across kingdoms and helpful in-depth functional mechanism analysis on these proteins.

**Abstract:**

Macrophage migration inhibitory factor (MIF), named for its role in inhibiting macrophage/monocyte migration, has multiple functions in modulation of inflammation, cell proliferation, angiogenesis, and tumorigenesis in vertebrates. Although homologs of this gene can be found in plants, the function of MIF in plants remains obscure. Here, we characterized *Ta*MIF1 in *Triticum aestivum* resembling the MIF secreted from *Homo sapiens*. Transcript analysis revealed that *TaMIF1* responded to stripe rust infection of wheat and was upregulated during the infection stage. *Ta*MIF1 was localized to both the cytosol and nuclei in wheat mesophyll protoplast. Additionally, *Ta*MIF1 possessed significant tautomerase activity, indicating conservation of MIFs across kingdoms. *Agrobacterium tumefaciens* infiltration assay demonstrated that *Ta*MIF1 was capable of suppressing programmed cell death hinting its role in plant immunity. Heterologous expression of *Ta*MIF1 increased fission yeast sensitivity to oxidative stress. Silencing *TaMIF1* decreased the susceptibility of wheat to *Pst* seemingly through increasing reactive oxygen species accumulation. In conclusion, functions of the *Ta*MIF1 were investigated in this study, which provides significant insight into understanding the role of MIFs across kingdoms.

## 1. Introduction

Macrophage migration inhibitory factor (MIF) is firstly identified from the culture supernatant of activated T lymphocytes, with the role of inhibiting the migration of macrophages [1]. In vertebrates, MIF is not only expressed within the cytosol of immune cells, but also expressed in various non-immune cells during the regulation of cellular functions [2,3,4]. MIFs in vertebrates were investigated as a pro-inflammatory cytokine regulating angiogenesis, apoptosis, and fibrosis [5,6], and identified to modulate innate and adaptive immune responses [7,8]. In addition, studies have shown that MIF can also be imitated or exploited by viruses and parasites to promote infection. Research progress in human medicine showed MIF is mainly involved in some inflammatory related diseases such as septic shock, rheumatoid arthritis, and even carcinoma [9,10]. Thus, multiple functions identified on MIF make it known as the “most interesting factor” in vertebrates [11].

Many breakthroughs have been investigated in MIF molecular mechanisms in vertebrates. *Hs*MIF was analyzed to have the significant tautomerase and oxidoreductase activity, which implies the role of MIF in cell cycle regulation. MIF has also been investigated as mediating the regulation of CD74-dependent MAPK signaling and activating cytosolic phospholipase A2 (cPLA2) [12,13], modulating tumor growth by interacting with c-Jun activation domain-binding protein-1 (JAB1) [14,15], and suppressing apoptosis associated with p53 activation and redox stress [16,17]. Recently, MIF was identified to be mediated in the interaction between the NOD-like receptor (NLRP3) and the intermediate filament protein vimentin [18]. All the above research results indicate that, in addition to its function as a cytokine, MIF seems to function through protein–protein interactions.

Some MIF-like proteins were identified in invertebrate species such as nematodes, ticks, and aphids. Their functions are almost involved in immune escape from hosts [19,20,21]. As an evolutionarily ancient molecule, multiple copies of *MIF* homologs have also been found in all higher plants [22]. Three *Arabidopsis thaliana* MIF-like proteins were firstly identified and in silico analysis, revealing that *At*MIF proteins resemble *Hs*MIF concerning predicted functional motifs [22]. This finding suggests some evolutionarily conserved features and biochemical function for MIFs across kingdoms. However, owing to the differences in immune responses between vertebrates and plants, the functions of MIFs in plants are still unclear. We noticed that in addition to the model plant *A. thaliana*, the *MIF* gene also exists in gramineous crops. As is well known, wheat is one of the world’s three major food crops, providing the calories for 20% of the world’s population [23]. On the other hand, wheat stripe rust (*Puccinia striiformis* f. sp. *Tritici*, *Pst*) has a wide range, high frequency, and causes serious damage, seriously threatening global wheat production and food security [24,25]. Therefore, research on the mechanism of wheat–*Pst* interaction provides a basis for the rational utilization of wheat resistance to stripe rust and the sustainable control of stripe rust.

Ma et al. [26] constructed an EST library for molecular mechanism investigation of wheat–*Pst* interaction. Interestingly, on the basis of this cDNA library, we obtain an EST sequence with an ORF with the characteristic MIF domain. Therefore, we aimed to explore the function of MIF in wheat, and to identify whether MIF is involved in the process of disease resistance in plants, especially whether wheat MIF is involved in the interaction between wheat and stripe rust. Here, we characterized TaMIF1 in *T. aestivum* resembling the MIF secreted from *H. sapiens*. Transcriptional analysis revealed that *TaMIF1* was upregulated in wheat leaves, regardless of whether they were challenged with the virulent *Pst* isolate CYR31 or the avirulent *Pst* isolate CYR23. The expression of the *Ta*MIF1–GFP fusion protein suggested that *Ta*MIF1 was localized to both the cytosol and nuclei in wheat mesophyll protoplast. Silencing *TaMIF1* decreased the susceptibility of wheat to *Pst* and increased ROS accumulation in wheat leaves. Our results indicate that *Ta*MIF1 seemingly negatively regulates the wheat immune system to *Pst* infection through the repression of ROS accumulation at the infection site, thereby promoting *Pst* infection.

## 2. Materials and Methods

### 2.1. Plant Materials, Fungal Isolates, and Bacterial Strains

Wheat cultivar Suwon 11 and *Pst* isolate CYR23 form the incompatible interaction, and Suwon11–CYR31 form the compatible interaction. Seeds of *N. benthemiana* were from the laboratory collection. Wheat protoplast was prepared using Suwon11 seedlings at two-leaf stage. CYR31 and CYR23 were grown and collected from wheat cultivar Suwon11 and Mingxian169, respectively [27]. *Escherichia coli* strain BL21 was used for prokaryotic expression system. Fission yeast isolate *Schizosaccharomyces*
*pombe* and *A. tumefaciens* strain GV3101 were used for gene expression.

### 2.2. Plasmid Constructs

Primers used throughout are found in the Appendix A. For the subcellular localization of *Ta*MIF1 assay, ORF of *TaMIF1* was cloned into the vector pTF486 [28]. PVX vector pGR107 was used to heterogeneously overexpress TaMIF1 in *N. benthamiana* [29]. Vector pREP3X was used for overexpression of TaMIF1 in fission yeast cells [30]. TaMIF1 was constructed in vector pGEX-4T-1 for prokaryotic expression. The BSMV vector was used for *TaMIF1* silencing [31].

### 2.3. Sequence Analysis and Domain Prediction

EST sequence of *TaMIF1* was from the cDNA library of wheat–*Pst* [26]. The obtained cDNA was cloned and aligned with the wheat genome database at EnsemblPlants (http://plants.ensembl.org/index.html accessed on 18 January 2021). All nucleotide and amino acid sequences were aligned using DNAMAN6. The expression profiles of TaMIF genes in different development stages were determined on the basis of the RNAseq data available from WheatOmics 1.0 (http://202.194.139.32/ accessed on 8 May 2021). The amino acid sequence of MIF from *H. sapiens* (accession no. NP_002406) was obtained from GenBank (http://www.ncbi.nlm.nih.gov accessed on 12 December 2020). The amino acid sequences of the three MIFs from *A. thaliana* were obtained from TAIR (http://www.arabidopsis.org/ accessed on 12 December 2020). Characteristic MIF domains were predicted with InterProScan 5 (http://www.ebi.ac.uk/interpro/ accessed on 12 December 2020). Nuclear export signal was predicted in NetNES 1.1 (http://www.cbs.dtu.dk/services/NetNES/ accessed on 12 December 2020).

### 2.4. RNA Extraction and qRT-PCR

Total RNAs were extracted using an RNAeasy R Plant Mini Kit (Qiagen, Shenzhen, China). The first-strand cDNA was synthesized using the Go Script Reverse Transcription System (Promega, Madison, WI, USA). Transcript profiles were measured using qRT-PCR. *TaEF-1* (*T. aestivum* elongation factor gene with accession no. Q03033) was used for normalization. A CFX Connect real-time PCR system (Bio-Rad, Singapore) was used to perform qRT-PCR, and the transcript levels were calculated using the 2^−ΔΔCT^ method [32].

### 2.5. Subcellular Localization of TaMIF1 in Wheat Protoplast

The wheat protoplast preparation refers to the protocol used for *A. thaliana* [33]. The plasmids carrying TaMIF1–GFP was transformed into wheat protoplast and incubated for 18 h in the dark. The fluorescence of transformed protoplasts was checked using the fluorescence microscope (Olympus Corporation, Tokyo, Japan) with the 485 nm laser line. Total protein was isolated from wheat protoplasts as previously described [34]. Western blotting was used for verifying protein expression. Detecting antibodies of mouse-derived GFP antibody and horseradish peroxidase-conjugated anti-mouse secondary antibody were all from commercial purchase (Sungene, Tianjin, China). The chemiluminescence substrate Pierce™ ECL Western Blotting Substrate was used for detection (Thermo Fisher Scientific, Waltham, MA, USA).

### 2.6. Heterogeneous Agroinfiltration Assays

Strain GV3101 carrying the expression plasmids were cultured to the late log phase. The infiltration buffer was used to resuspend cells to an OD600 of 0.2. Before infiltration, cells needed to be incubated in the dark for 2 h. Cell suspension carrying *Ta*MIF1 and Bax were infiltrated into the tobacco leaves one day apart. Photos were taken after 3–5 days’ infiltration. In order to degrade chlorophyll, we decolorized the leaves using ethanol. eGFP was used as the negative control. The positive control was the oomycete elicitor Avr1b that suppressed Bax-induced PCD [35]. Western blotting was used to detect protein expression.

### 2.7. Tautomerase Activity Assay

pGEX-4T-TaMIF1 was expressed in *E. coli* strain BL21. Proteins were induced and collected after incubating at LB medium with 1.0 mM IPTG overnight at 16 °C. Cells were collected and lysed by sonication. After centrifugation, soluble proteins were in the supernatant and analyzed by SDS-PAGE. For further tautomerase activity assay, GSTrap was used to purify the fusion proteins. Tautomerase activity was measured using L-dopachrome methyl ester [36,37], which was prepared just before use by adding L-3,4-dihydroxuphenylalanine methyl ester (4 mM) and sodium periodate (8 mM) to a reaction buffer (1 mM EDTA, pH 6.2). The mixture was incubated for 5 min at room temperature and placed on ice for 20 min. The activity was determined at room temperature by adding 800 μL of the prepared L-dopachrome methyl ester to a cuvette containing *Ta*MIF1 or *Mm*MIF expressed from a prokaryotic expression system and measuring the decrease in absorbance at 475 nm for 15 min. The rate decrease of absorbance was calculated after 60 s. Empty vector pGEX-4T was used as negative control and *Mm*MIF was used as the positive control.

### 2.8. The Heterogenous Overexpression in Fission Yeast Assay

pREP3X-TaMIF1 was conducted and transformed into *S. pombe* by electroporation. Thiamine was the repressor that can inhibit promoter of pREP3X. Transformed cells diluted to OD600 = 0.2 were cultured in inducing (−Thiamine) or repressing (+Thiamine) medium for about 20 h at 30 °C in the dark. Then, cells were collected by centrifugation; washed with ddH_2_O; and finally diluted to densities of 10^6^, 10^5^, 10^4^, and 10^3^ cell mL^−1^ and removed on yeast medium (adding 0.5 mM H_2_O_2_ for H_2_O_2_ sensitivity assay and adding 0.35 M NaCl for hyperosmotic stress assay). The morphology of yeast cells was observed using the microscope in brightfield.

### 2.9. Tranient Gene Silencing of TaMIF1 in Wheat cv. Suwon11

Virus-induced gene silencing was carried out as described [31,38]. BSMV:TaMIF-1as and BSMV:TaMIF-2as were constructed and used for inoculating the second leaves of wheat seedlings. BSMV:TaPDS-as and BSMV:00 were used as controls. Ten days after viral inoculation, the fourth leaves of wheat seedlings were inoculated with stripe rust of fresh CYR31 urediospores. Phenotypes of the fourth leaves were observed and photographed 14 days after *Pst* inoculation. *Pst*-challenged fourth leaves of wheat seedlings inoculated with BSMV in advance were collected for RNA extraction at 0, 12, 24, 48, and 120 hpi. qRT-PCR was used to evaluate the silencing efficiencies of *TaMIF1* using the RNA extracted at each time point.

### 2.10. Histological Sample Treatment and Observation

DAB staining was used to observe H_2_O_2_ accumulation [39]. Virus-inoculated leaves were infected with CYR31 and sampled at 12, 24, and 48 hpi. Leaf segments were treated in ethanol/acetic acid (1:1, v/v). Samples were immersed in DAB solution for 8 h under light at room temperature. Infection structures of *Pst* were stained with wheat germ agglutinin conjugated to Alexa 488 (WGA488) (Invitrogen, Carlsbad, CA, USA) [40]. Virus-inoculated leaves were sampled at 24 and 120 hpi with CYR31 infection. The decolorized samples were autoclaved in 1M KOH for 2 min, washed twice with 50 mM Tris (pH 7.5), and stained with WGA488 (20 μg/mL) for 30 min. The stained samples were viewed with the fluorescence microscope and measured using DP-BSW software (Olympus Corporation).

## 3. Results

### 3.1. Genome-Wide Identification of MIF-Like Genes in Wheat and Their Expression

On the basis of the EST sequence WRIS 3105 from the cDNA library of *Pst*–wheat interaction [26], a MIF-like gene with an ORF of 348 bp was identified and cloned from the wheat cultivar Suwon11. Sequence analysis revealed that the isolated wheat MIF gene (designated as TaMIF1) has three copies located on wheat chromosomes 5A, 5B, and 5D. Another paraologuous MIF gene (designated as TaMIF2) was also searched in the wheat genome, being found to be located on chromosomes 7A, 7B, and 7D. The nucleotide sequences of the two group genes were found to be highly similar (82.85%), making it difficult to study individual *TaMIF* expression profiles. However, according to transcription analysis, homologous genes on chromosomes 5D and 5B responded to the induction of *Pst*, and homologs on 5A, 7A, 7B, and 7D were not induced (Figure 1A). Thus, we chose *TaMIF1* as a representative to carry out further research. In particular, we found that *TaMIF1* has two transcripts. To identify expression profiles of different transcripts in wheat and *Pst* interaction, we designed specific primers for qRT-PCR (Appendix A). The results in Figure 1B indicate that the two transcripts have similar expression profiles. The transcript levels are both induced during *Pst* infection (Figure 1B).

Through sequence alignment and domain prediction, we found that wheat MIF proteins have similar domains to *Hs*MIF and MIFs in *A. thaliana*. Amino acid sequence alignment of *Hs*MIF, *At*MIFs, and *Ta*MIFs indicates that they all contain three conserved residues (Pro1, Lys32, and Ile64) of the active sites for tautomerase activity identified in *Hs*MIF (Figure 1C, black triangle), but plant MIFs lack the C-X-X-C motif that conducts the oxidoreductase activity. Our results suggest that *Ta*MIFs are phylogenetically ancient proteins that seem evolutionarily conserved.

### 3.2. TaMIF1 Showed Tautomerase Activity

With the prediction by ExPASy, a tautomerase domain was found in *Ta*MIF1 as all MIFs reported. Thus, the protein of *Ta*MIF1 was expressed in a prokaryotic expression system to investigate its enzymatic activity (Appendix A). In this study, *Mm*MIF, the MIF from *Mus musculus*, was used as a positive control. As shown by the results, with L-dopachrome methyl ester as the substrate, *Ta*MIF1 and *Mm*MIF displayed significant tautomerase activity as assessed by the decreasing absorbance at 475 nm (Figure 2). This result suggests conservation between *Ta*MIF1 and MIFs from vertebrates in the tautomerase activity.

### 3.3. TaMIF1 Was Localized to the Cytosol and the Nuclei

In vertebrates, most MIFs are predominantly stored in cytosols [5], and they could also be detected in small vesicles [41] and the nucleus of vertebrate cells [42]. To identify the subcellular localization of *Ta*MIF1 in wheat, we performed a transient expression of *Ta*MIF1–GFP fusion protein in wheat mesophyll protoplasts. Fluorescence of the fusion protein was observed in both the cytoplasm and the nucleus, as was fluorescence of the control expressing GFP coding sequence only (Figure 3). Expression of GFP and the *Ta*MIF1–GFP fusion were confirmed by Western blot analysis (Appendix A). These results indicate that *Ta*MIF1 localizes to both the cytoplasm and nucleus in wheat cells.

### 3.4. Transient Expression of TaMIF1 in Tobacco Suppressed Programmed Cell Death

In vertebrates, MIFs are important regulators of cell apoptosis, which sustains macrophage survival and suppresses p53-dependent apoptosis in vertebrates [17,43]. It is interesting to note that the BCL2-associated X (Bax) protein from mammals can trigger programmed cell death (PCD) when expressed in plants from a tobacco mosaic virus vector (PVX) [44]. To determine whether *Ta*MIF1 is related to PCD, we overexpressed *Ta*MIF1 in *N. benthamiana* using the PVX vector. When leaves were infiltrated with eGFP (negative control), Avr1b (positive control) [35], or *Ta*MIF1, no cell death was observed. The site inoculated with eGFP then challenged by *Agrobacterium* expressing Bax 24 h later exhibited obvious cell death, while the sites inoculated with *Agrobacterium* expressing Avr1b or *Ta*MIF1 then challenged with Bax showed no cell death, indicating that the cell death induced by Bax was successfully suppressed by Avr1b or *Ta*MIF1 (Figure 4). Western blotting of proteins expression are shown in Appendix A. The result indicates that *Ta*MIF1 suppresses PCD directly in plants.

### 3.5. Overexpression of TaMIF1 in the Fission Yeast Affected Cell Morphology and Increased the Sensitivity to H_2_O_2_

In vertebrates, MIFs play an important role in cell proliferation. To investigate whether *Ta*MIF1 also has a role in cell growth and sensible to some environmental stimuli, we constructed *TaMIF1* to the pREP3X vector [30] and transformed pREP3X-*Ta*MIF1 into *S**. pombe*
*cells*. Yeast cells expressing pREP3X- and pREP3X-*Ta*MIF1 were cultured in inducing (−Thiamine) or repressing (+Thiamine) medium, respectively. Although the growth of yeast did not change significantly, microscopic observation revealed that the morphology of yeast cells transformed with pREP3X-*Ta*MIF1 changed from a long strip to a round shape, showing an aggregated state (Figure 5A). The cell morphology of all transformed yeast appeared to be normal. The statistical results showed that cell length was significantly shorter compared with the cells grown in the repressing medium (Figure 5B). Therefore, overexpression of *Ta*MIF1 can affect yeast cell length.

We further examined the effects of *Ta*MIF1 on the survival of yeast cells subjected to H_2_O_2_ (oxidative stress) and NaCl (hyperosmotic stress). As shown in Figure 5C, the viability was severely reduced in the *Ta*MIF1-transformed yeast cells grown on an inducing medium with H_2_O_2_ compared with controls. However, no difference appeared in the viability of yeast cells grown on a medium with NaCl (Figure 5C). The results indicate yeast cells are more sensible to H_2_O_2_ when *Ta*MIF1 exogenously overexpressed in yeast cells, which hints that TaMIF1 may be involved in the regulation of reactive oxygen species.

### 3.6. Silencing TaMIF1 Expression Reduced Wheat Susceptibility to Pst

To characterize the role of *Ta*MIF1 in wheat-*Pst* interaction, we applied barley stripe mosaic virus (BSMV)-mediated gene silencing [31]. Two different fragments were chosen to specifically silence *TaMIF1* (Figure 6A). Then, all the possible targets of the two fragments were predicted with the RNAi off-target prediction tool SI-FI 3.2 in both the genome of wheat and *Pst* (Appendix A). At 10 d after inoculation with BSMV, chlorosis was observed in wheat leaves and obvious photobleaching was observed in plants inoculated with BSMV:TaPDSas (*Ta*PDS, wheat phytoene desaturase) (Figure 6B), indicating that the RNAi system was effective. BSMV-inoculated wheat plants were then infected with CYR31 to show effects of *Ta*MIF1 in the *Pst*–wheat interaction. In the *Pst*–wheat interaction system, both BSMV:TaMIF-1as and BSMV:TaMIF-2as inoculated plants showed delayed and decreased pustules formation (Figure 6C,D). Transcript levels of *TaMIF1* in *TaMIF1*-silenced plants were significantly reduced using either of the two fragments (Figure 6E).

On the basis of the altered susceptibility phenotype of *TaMIF1*-silenced wheat plants, we assayed cytological changes in VIGS-silenced plants inoculated with *Pst*. Histological observations of *Pst* growth and development in wheat plants are shown in Appendix A and Figure 7. Statistical analysis shows that the lengths of infective hyphae at 24 hpi and the size of the infected area at 120 hpi decreased significantly in the compatible interaction when *TaMIF1* was silenced (Figure 7). Thus, silencing *TaMIF1* appears to be beneficial for fungal invasion of wheat in the compatible interaction, and wheat seedlings are less susceptible to *Pst* with *TaMIF1* silenced.

### 3.7. Accumulation of ROS Was Induced in TaMIF1-Silenced Plants

Since silencing *TaMIF1* decreases the susceptibility of wheat to *Pst*, we assessed the plant immune responses in *TaMIF1*-silenced plants challenged with *Pst*. ROS is an important marker of host resistance response to pathogen and is usually generated at the infection unit [38,45]. As shown in Figure 7A, H_2_O_2_ accumulation mainly occurred in the guard cells in the early stage of 12 hpi, and the H_2_O_2_ amount in *TaMIF1*-knockdown plants was not affected. Significant changes were observed at 24 and 48 hpi. The accumulation of H_2_O_2_ per infection site was significantly induced in *TaMIF1*-silenced plants, no matter whether in guard cells or in attacked mesophyll cells (Figure 8). These results suggest that silencing *TaMIF1* induces ROS accumulation in *Pst*-infected wheat plants and indicates that *Ta*MIF1 might negatively affect the innate immune system of wheat plants.

## 4. Discussion

As one of the first cytokines described, MIF emerged as a pivotal regulator of innate immune systems in vertebrates [5,8]. MIF-like proteins identified in invertebrate species are almost involved in immune escape from hosts [19,20,21]. As a phylogenetically ancient protein, MIFs were also found in almost plant species [22]. Nevertheless, the function of MIFs in plants remains unknown. In this study, we characterized functions of *TaMIF*1 gene in wheat. Our results show that TaMIF1 acted as a negative immunity regulator of wheat resistance against *Pst*, suggesting a conservation function in the innate immune system of animals and plants.

Unlike the single MIF copy found in vertebrates, more than one MIF copies were identified in higher plants, which hinted that different MIF copies may exhibit different functions with different expression patterns [22]. There are two MIF genes in wheat, *TaMIF1* with three copies located on chromosomes 5A, 5B, 5D, and *TaMIF2* with three copies located on chromosomes 7A, 7B, 7D. According to transcription analysis from the wheat expression database, only *MIF*s on chromosome 5B and 5D are upregulated during wheat–*Pst* interaction. Although the similarity level of *Ta*MIFs to *Hs*MIF is only 34.48%, the domains and structure of the proteins seem highly conserved. Combined with the enzymatic activity as a tautomerase found for *Ta*MIF1, we conclude that MIFs are not only conserved in sequence, but also have an overlap in function across kingdoms. MIFs have been subject to various mimicry mechanisms exploited by viruses and parasites and involved in the pathogenesis of septic shock, arthritis, other inflammatory conditions, and even carcinoma [9,10]. In addition, the plant immune responses could be repressed by aphid-secreted MIFs [20]. The function of plant MIFs in plant immune system seems much more intriguing. Transcription analysis showed that *TaMIF1* could respond to *Pst* infection, which hinted that *TaMIF1* participates in wheat–*Pst* interaction.

In vertebrates, MIFs showed multiple expression locations, being constitutively expressed not only in immune system cells but also in epithelial cells that are in direct contact with the host’s natural environment [5]. As for subcellular localization, although most MIFs are predominantly stored in cytosols [5], they could also be detected in small vesicles [41] and the nucleus of vertebrate cells [42]. In this study, the expression of *Ta*MIF1–GFP fusion protein suggested that *Ta*MIF1 localized in both the cytoplasm and the nucleus. The only former localization result of MIFs in *A. thaliana* also localized in the cytosol and the nucleus [22]. This is following the localization of MIFs in vertebrates. This result also suggested the conservation in the localization of MIFs in both vertebrates and plants. In addition, the localization of *Ta*MIF1 might imply some multiple biological functions that are consistent with those in vertebrates [46].

Bax, a member of the Bcl-2 family, triggers cell death in animals [47], while in vertebrates, MIFs are important regulators of cell apoptosis and can suppress Bax-induced cell death [17,48]. It also has been identified that when expressed in plants, Bax can also activate an endogenous cell death program that is very similar to the programmed cell death of plants caused by pathogen infection [44]. In this study, we showed that *Ta*MIF1 could suppress PCD induced by the Bax gene in *N. benthamiana,* which indicated that MIFs both in animals and plants share the same function of suppressing cell death. In both plants and animals, PCD fulfills the same role of eliminating the diseased cells [49,50]. Thus, the results also hints that *Ta*MIF1 may have a role in plant immunity to pathogens. In this study, silencing of *TaMIF1* enhanced immunity to *Pst* in *TaMIF1*-knockdown plants. In response to the virulent *Pst* CYR31, *TaMIF1*-knockdown plants were less susceptible with delayed and decreased sporulation. Histological observations revealed silencing of *TaMIF1* led to increased H_2_O_2_ accumulation at neighboring mesophyll cells and infected sites. The induced *TaMIF1* expression at 24 and 48 hpi in compatible interaction may be responsible for the suppression of H_2_O_2_ production. The increase of host-produced ROS (H_2_O_2_) could be indirect as a result of increased production/secretion of a defense-triggering host response. Moreover, overexpression of *Ta*MIF1 can affect cell morphology, and yeast cells were found to be more sensible to H_2_O_2_, hinting that *Ta*MIF1 may be involved in the regulation of ROS. Thus, we concluded that *Ta*MIF1 might negatively regulate the plant immune system to *Pst* infection through the repression of ROS accumulation at the infection site, thereby promoting *Pst* infection. However, the mechanism of *Ta*MIF1 regulating ROS remains to be investigated.

## 5. Conclusions

The present study showed that MIFs are important endocrine immune factors in vertebrates. MIF homologs have also been found in all higher plants, but studies on the role of MIFs in plants are limited. In this study, a function of MIFs (*Ta*MIF1) in the wheat innate immune system was uncovered for the first time. It could be speculated that MIF is involved in pathogen–host interaction in both vertebrates and plants. This study demonstrated the functions of *Ta*MIF1 in tautomerase activity and suppressing PCD, which are consistent with MIF in vertebrates. *Ta*MIF1 was shown to be localized in both the cytosol and nuclei in wheat protoplast. In addition, we investigated that *Ta*MIF1 is upregulated in the process of wheat–*Pst* interaction and *Ta*MIF1 functions as a negative regulator in wheat resistance to *Pst* dependent on limiting ROS release. The proposed results hinted that TaMIF1 mainly modulates the ROS signaling, and then alters the subsequent immune responses. Nevertheless, the molecular mechanism of *Ta*MIF1 is still unclear and requires further exploration. In the future, our work should focus on seeking the possible targets of *Ta*MIF1 that can help to explore its possible role in wheat immunity. Further, because *Hs*MIF has multiple functions in a variety of biological processes [10], other functions of MIFs in plants deserve further studies. The function characterization of wheat MIF in this study should aid in understanding the biological role of MIFs across kingdoms and be helpful for future in depth functional mechanism analysis on these proteins.

## Figures and Tables

**Figure 1 biology-10-00878-f001:**
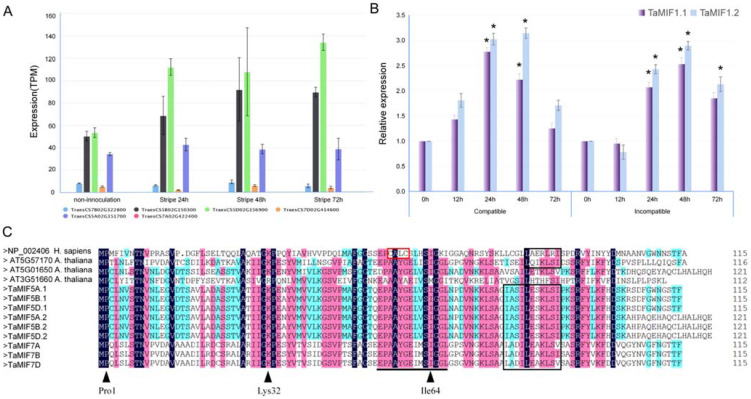
Sequences analysis of *Ta*MIFs. (**A**) Developmental expression profile of *Ta*MIFs. Expression analysis of TaMIF genes in different *Pst* infection periods; column height shows expression of *TaMIF* genes on the basis of log-transformed TPM (transcripts per million) value. (**B**) Transcript profiles of two transcripts on wheat chromosome 5B. Relative transcript levels were analyzed in both compatible and incompatible wheat–*Pst* interactions from 0 hpi to 72 hpi. Relative transcript levels of *TaMIFs* were calculated by the comparative 2^−ΔΔCt^ method with *TaEF1* as internal standard and relative to that of 0 hpi. Bars indicate means of three independent biological replicates (±SE). Asterisks indicate significant differences (*p* < 0.05) relative to the 0 hpi sample. (**C**) Multi-sequence alignment of *Hs*MIF, *At*MIFs, and *Ta*MIFs. Amino acid sequences of *Hs*MIF, *At*MIFs, and *Ta*MIFs were aligned with DNAMAN 6.0 using standard parameters. Identical amino acids are shown in black; black triangles are tautomerase active site residues that exist in MIFs; red box represents the C-X-X-C motif in *Hs*MIF. The black line indicates the location of InterProScan domain IPRO19829 (MIF, conserved site) in MIFs; black boxes indicate the predicted NES.

**Figure 2 biology-10-00878-f002:**
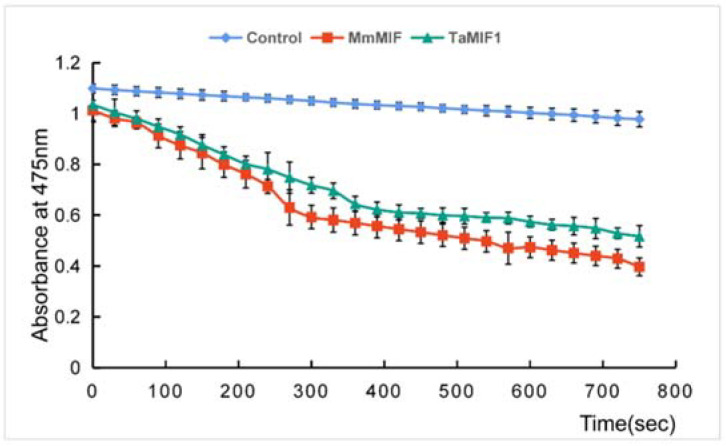
The tautomerase activity of *Ta*MIF1. The empty vector was used as a negative control, and *Mm*MIF, the MIF from *Mus musculus*, was used as a positive control. The results showed means ± SD of four independent replicates.

**Figure 3 biology-10-00878-f003:**
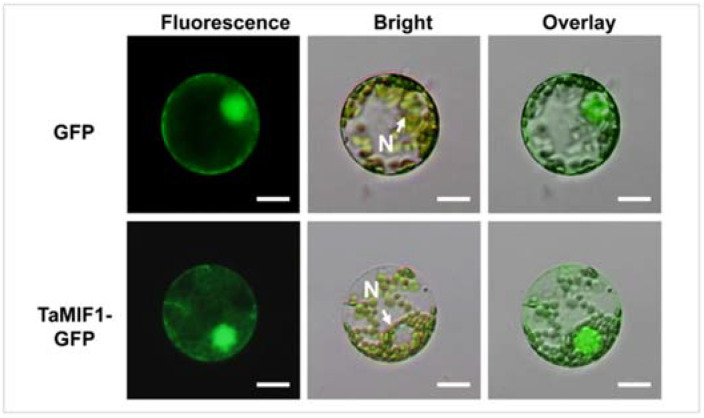
Subcellular localization of TaMIF1. Subcellular localization of TaMIF1–GFP was observed in wheat mesophyll protoplasts with GFP as control. Overlay is a combination of both fluorescence and bright field. N: nucleus. Bars = 50 μm.

**Figure 4 biology-10-00878-f004:**
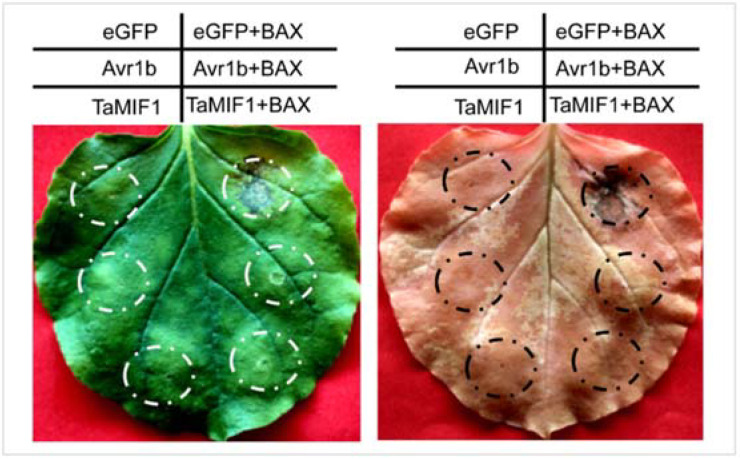
Heterogeneous expression of TaMIF1 in *N. benthamiana* suppressed Bax-induced PCD. *A. tumefaciens* cells containing PVX:TaMIF1 were infiltrated into *N. benthamiana* leaves. eGFP and Avr1b were used as negative and positive controls, respectively. After infiltrating for 24 h, *A. tumefaciens* cells carrying PVX:Bax were infiltrated at the same site. Photos were taken 5 days after infiltration, and the leaves were decolorized with ethanol.

**Figure 5 biology-10-00878-f005:**
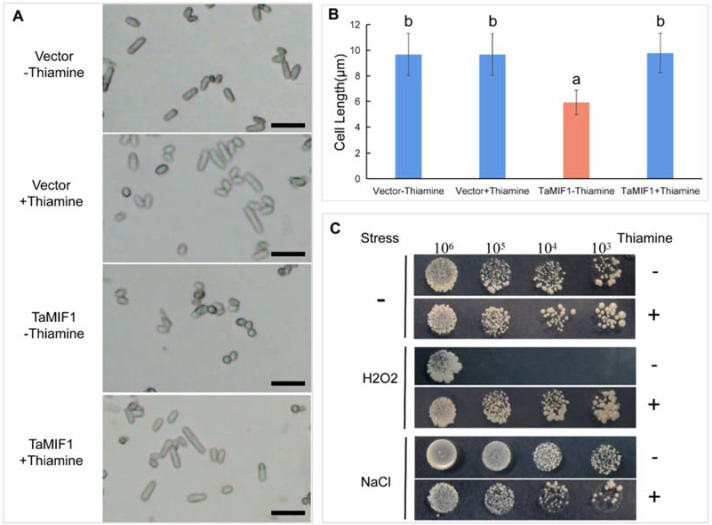
Effects of overexpressing *Ta*MIF1 in yeast. (**A**) TaMIF1 expressed in yeast could affect cell length. Yeast cells were observed under a microscope in bright field. Bars = 20 μm. (**B**) The length of yeast cells were calculated. Differences were assessed using ANOVA with results from three replicates. (**C**) TaMIF1 expressed in yeast increased sensibility of yeast cells to oxidative stress. Yeast cells expressing TaMIF1 were grown on yeast medium containing 0.5 mM H_2_O_2_ or 0.35 M NaCl.

**Figure 6 biology-10-00878-f006:**
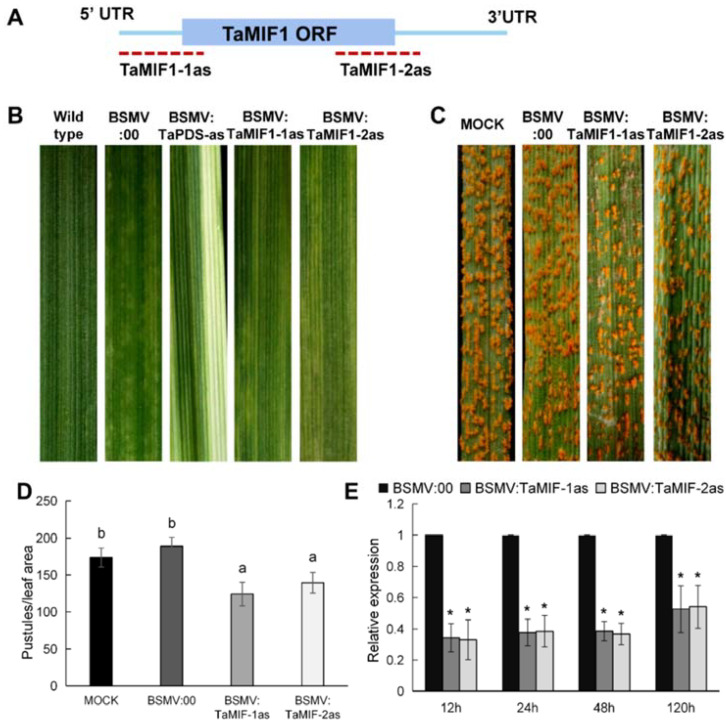
Silencing *TaMIF1* decreased susceptibility of wheat to *Pst.* (**A**) Red dotted lines represent the two specific regions used for silencing *TaMIF1*. (**B**) Phenotypes of leaves inoculating with BSMV. Chlorotic virus symptoms were observed on the fourth leaves of wheat seedlings inoculated with either BSMV:00, BSMV:TaMIF1-1as, or BSMV:TaMIF1-2as. Obvious photo bleaching was observed in wheat plants inoculated with BSMV:TaPDS-as. (**C**) Delayed and decreased *Pst* pustule formation in *TaMIF1*-silenced wheat leaves. (**D**) The quantitative data of pustules in *Pst*-infected wheat plants. Differences were assessed using ANOVA with results from three replicates. (**E**) Relative transcript levels of *TaMIF1* in *TaMIF1*-silenced plants in the *Pst*–wheat interaction. Values were expressed relative to *TaEF1*, with the empty vector (BSMV:00) set as 1. Bars represent means (±SE) from three biological replicates. Asterisks indicate significant differences (*p* < 0.05).

**Figure 7 biology-10-00878-f007:**
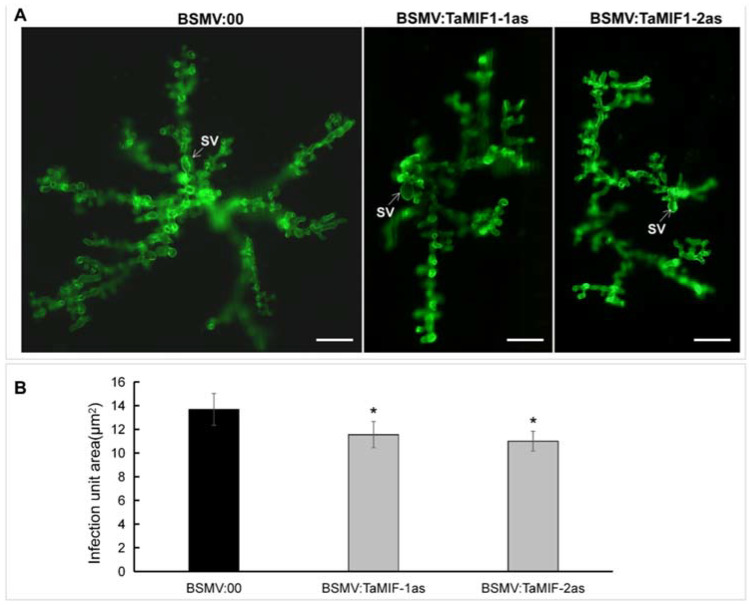
Fungal growth of CYR31 in wheat leaves inoculated with BSMV:00 or BSMV:TaMIF1 at 120 hpi. (**A**) The micrograph of the fungal structure stained by WGA in wheat at 120 hpi. Bars = 50 μm. (**B**) The statistical data of infected area in wheat at 120 hpi. Results were obtained from 50 infection sites and values represent mean ± SE of three independent replicates. Differences were assessed using Student’s *t*-tests. Asterisks indicate *p* < 0.05.

**Figure 8 biology-10-00878-f008:**
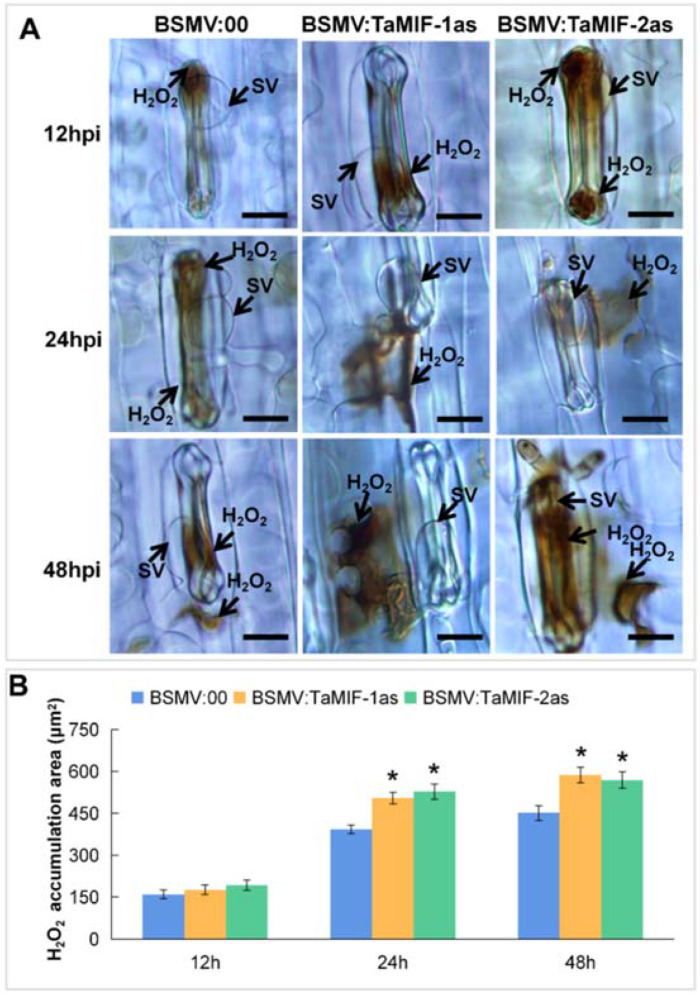
H_2_O_2_ accumulation in *TaMIF1*-knockdown wheat plants challenged with *Pst.* (**A**) Accumulation of H_2_O_2_ was monitored with DAB staining at 12, 24, and 48 hpi in a compatible interaction of *Pst* and wheat with *TaMIF1* silenced or not. Bars = 20 μm. SV, sub-stomatal vesicle. (**B**) Statistical analysis of H_2_O_2_ accumulation in *Pst*–wheat interaction with *TaMIF1* silenced or not. All mean values were obtained from three biological replicates with 50 infection sites per replicates. Asterisks indicate significant differences (*p* < 0.05).

## Data Availability

The relevant datasets supporting the results of this article are included within the article and its additional files.

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
