# Peer review of "Functional Characterization of the Wheat Macrophage Migration Inhibitory Factor TaMIF1 in Wheat-Stripe Rust (Puccinia striiformis) Interaction"

_biology, 2021, doi:10.3390/biology10090878_

Round 1

Reviewer 1 Report

Thank you for the opportunity to review the manuscript tited, Functional characterization of the wheat macrophage migration inhibitory factor TaMIF1 in wheat-stripe rust (Puccinia striiformis) interaction.

I feel this research is interesting. However, I have provided some minor comments below.

  1. Could you please elaborate more on the Introduction part focusing more on challenges/gaps and objectives you are trying to achieve?
  2. I would also write more on the conclusion and future work. 
  3. Please introduce all acronyms. Also, there are several linguistic and grammatical issues such as typos, spacing error, font size that needs to be corrected. 

Author Response

1.Could you please elaborate more on the Introduction part focusing more on challenges/gaps and objectives you are trying to achieve?

Response: Thanks for your comment. Considering your suggestions in depth, we emphatically added challenges/gaps and objectives we are trying to achieve in the introduction section. Please see line 73-95 of the manuscript for the revised content.

2.I would also write more on the conclusion and future work. 

Response: Thank you for your suggestions. In the revised manuscript, we have enriched the conclusion of the article and added more future work prospects. Please see line 420-439 of the manuscript for the revised content.

3.Please introduce all acronyms. Also, there are several linguistic and grammatical issues such as typos, spacing error, font size that needs to be corrected. 

Response: Thank you for your comments. We have added abbreviations at the end of the article to explain all acronyms that appear in the manuscript. Please see line 440-447 of the manuscript for the revised content. We have also checked and revised the linguistic and grammatical problems in the manuscript one by one. We believe there would be a great improve in the revised manuscript.

Reviewer 2 Report

The manuscript "Functional characterization of the wheat macrophage migration inhibitory factor TaMIF1 in wheat-stripe rust (Puccinia striiformis) interaction" studied the wheat TaMIF1 and TaMIF2 genes in the defense responses to Pst. The study was well designed, and the manuscript was well-written. The results were supported by appropriate experiments with methods clearly described, and the conclusion provided novel insight to the field of interest. The only concern that I would remind the authors is the sentence in the line 13-15 of the Simple Summary, and in the line 21-24 of the Abstract were exactly identical. This would raise the flag of self-plagiarism. It seems that the authors may re-write the simple summary to provide more concise and concentrated idea. The current simple summary contains descriptive and redundant idea, e.g. line 15-16 and line 18-19 basically says the same thing. Other than revision on the simple summary, the manuscript is ready for publication.  

Author Response

Thank you for your comments and good suggestion. We have rewritten the simple summary to better summarize and display the content of the manuscript. Please see line 13-23 of the manuscript for the revised content.

Reviewer 3 Report

This manuscript describes some potentially interesting findings related to a possible role for TaMIF1 in interactions with Pst.
The current manuscript lacks some important information and does not fully incorporate investigation of the three homoeologues (A, B D genome versions) in the study. The authors incorrectly refer to the homoeologues as 'alleles'. They are NOT alleles.
Throughout the manuscript the authors treat wheat as a diploid species and do not take into account the three homoeologues copies of the MIF genes. This must be rectified.
For example:
The authors show only a single protein sequence without describing whether it derives form the A, B or D genome. The remaining investigations do not determine whether the expression or silencing is affecting the three homoeologues equally.
All the relevant information is available in the wheat sequence in 'Ensembl Plants' and in the expression information in 'Wheat Expression' database. This needs to be incorporated into the manuscript and experiments designed to show homoeologue-specific effects.
I have attached a file showing the protein sequences of the homoeologues of the two genes and the two transcripts versions of the MIF2
The expression data shown in Figure 2 does not show which homoeologues are being assessed. Ensembl Plants shows the group 5 homoeologues to have two transcripts expressed. Which transcripts are being assessed in Figure 2?
The expression data in fig 2 is not generally supported by that present in 'Wheat Expression' database with the exception of the effect of PEG on the A genome homoeologue of MIT2 in one cultivar.
It is not possible to determine which homoeologue expression is being shown in Figure 2 as no information is provided on the primers used and their efficiency with the homoeologues.
What are the PCR primers used throughout and how do they function for the three homoeologues?
The heat map in Figure 1 (c) lacks values in almost all cells but the cells have different colours suggesting different levels of expression. This Figure is the only occasion in which the authors refer to the homoeologues of the group 5 and 7 genes.
Why do the authors not study MIF2 at all?
No quantitative data is provided to support the image shown in Fig 5
Some elements of the experimental design are not explained. No mention is given on the isolate of Pgt used. Is this virulent or avirulent on the cultivar. Why is this experiment included when all the rest of the manuscript (and the title) relates to Pst?

Author Response

We thanks for your careful read and thoughtful comments on previous draft. The revised manuscript has uploaded and we have provided a point-by-point response in a word file uploaded below. Please see the attachment.

Round 2

Reviewer 3 Report

Thank you for your positive response to my earlier comments. The new version is much improved. I have a few minor comments and a suggestion.

The 5A, 5B and 5D  genes are homoeologues of TaMIF1

The 7A, 7B and 7D genes are homooelogues of TaMIF2

The represent two independent (perhaps paraologuous) genes and should be kept separate in the text. The current wording sometimes conflates the two genes and infers six homoeologues.

I am a little concerned by the wording used in the Discussion on line 366. The authors state that the RNAseq data on expression of the homoeologues of TaMIF1 and TaMIF2 shown in fig 1A is 'our results'. Did the authors carry out these experiments of are they showing data obtained from the wheat expression database in Ensembl. I suspect that it is the latter in which case they should properly acknowledge in full the researchers who produced this data.

The emphasis of the text and title relate to affects of MIF1 on susceptibility to Pst. No quantitative data is presented in the manuscript to support the images in Fig. 6C.

I strongly urge the authors to move Supplementary figure 6 into the main text to provide quantitative data on the effect of silencing of TaMIF1 on susceptibility to Pst.

Author Response

Thanks for your valuable suggestion again. Here are my response to the Comments one by one:
Comments 1:
The 5A, 5B and 5D  genes are homoeologues of TaMIF1
The 7A, 7B and 7D genes are homooelogues of TaMIF2
The represent two independent (perhaps paraologuous) genes and should be kept separate in the text. The current wording sometimes conflates the two genes and infers six homoeologues.
Response1:  Thank you for your comments and good suggestion. We corrected our words in the revised manuscript to separate the two genes. Please see line 201-207 and line 379-383 of the manuscript for the revised content.
Comments 2:
I am a little concerned by the wording used in the Discussion on line 366. The authors state that the RNAseq data on expression of the homoeologues of TaMIF1 and TaMIF2 shown in fig 1A is 'our results'. Did the authors carry out these experiments of are they showing data obtained from the wheat expression database in Ensembl. I suspect that it is the latter in which case they should properly acknowledge in full the researchers who produced this data.
Response2: Thanks for your valuable advice about our negligence. We sincerely thank all the researchers who contributed to the wheat genome database the wheat expression database in Ensembl. We have rewritten this part in Discussion. We also showed our gratitude to the researchers in the acknowledgment section in the manuscript. Please see line 381-383 and line 465-466 of the manuscript for the revised content.
Comments 3:
The emphasis of the text and title relate to affects of MIF1 on susceptibility to Pst. No quantitative data is presented in the manuscript to support the images in Fig. 6C.
Response3: We are grateful for the suggestion. We added the statistical analysis of pustules in the wheat plants in Fig. 6D. The previous Fig. 6D was changed to Fig. 6E. Please see line 323-334 of the manuscript for the revised content.
Comments 4:
I strongly urge the authors to move Supplementary figure 6 into the main text to provide quantitative data on the effect of silencing of TaMIF1 on susceptibility to Pst.
Response4: Thank you for your kind suggestion. Considering your comment and to make the data more complete, we moved Supplementary figure 6 into the main text as Fig 7 to exhibit quantitative data on the effect of silencing of TaMIF1 on susceptibility to Pst. The previous Fig. 7 was changed to Fig. 8. Please see line 343-348 of the manuscript for the revised content.